# Open-channel microfluidics via resonant wireless power transfer

Christopher T. Ertsgaard [1], Daehan Yoo[1], Peter R. Christenson[1], Daniel J. Klemme[1] & Sang-Hyun Oh [1✉]

Open-channel microfluidics enables precise positioning and confinement of liquid volume to interface with tightly integrated optics, sensors, and circuit elements. Active actuation via electric fields can offer a reduced footprint compared to passive microfluidic ensembles and removes the burden of intricate mechanical assembly of enclosed systems. Typical systems actuate via manipulating surface wettability (i.e., electrowetting), which can render low-voltage but forfeits open-microchannel confinement. The dielectric polarization force is an alternative which can generate open liquid microchannels (sub-100 μm) but requires large operating voltages (50–200 $V_{RMS}$) and low conductivity solutions. Here we show actuation of microchannels as narrow as 1 μm using voltages as low as 0.5 $V_{RMS}$ for both deionized water and physiological buffer. This was achieved using resonant, nanoscale focusing of radio frequency power and an electrode geometry designed to abate surface tension. We demonstrate practical fluidic applications including open mixing, lateral-flow protein labeling, filtration, and viral transport for infrared biosensing—known to suffer strong absorption losses from enclosed channel material and water. This tube-free system is coupled with resonant wireless power transfer to remove all obstructing hardware — ideal for high-numerical-aperture microscopy. Wireless, smartphone-driven fluidics is presented to fully showcase the practical application of this technology.

[1] Department of Electrical and Computer Engineering, University of Minnesota, Minneapolis, MN 55455, USA. ✉email: sang@umn.edu

Microfluidics offers precise positioning and packing of liquid samples over ever-shrinking detectors, pixel displays, and optical pathways[1–4]. The process of conversion from the macro-scale of human handling to densely packed microchannel arrays can be cumbersome, and bulky tubing, pumps, and intricately fabricated interconnects remain the state-of-the-art (Fig. 1A). Electrowetting on dielectric (EWOD)—which actuates entire liquid droplets rather than narrow and confined channels, has gained broad interest within the scientific community, revealing a great desire for low-power electrofluidics[5]. Current applications span tunable optics[6–10], biological assays[11–16], microhydraulics[17], mechanical energy harvesting[18], solid–liquid-phase interaction studies[19], and digital displays[20–23]. Compared with microfluidics, however, EWOD is ill-suited for micron-precise fluid and analyte placement, high-throughput channel packing, and multi-flow-based actions (e.g., localized chemical reactions, mixing and/or filtering). Electro-actuated, capillary-driven microfluidics alternatively can confine sample solution within enclosed channels by using electric fields to wet and bridge the liquid to regions of high capillary action[24]. However, without the support from the channel enclosures, surface tension (ST) greatly impedes actuation of high-aspect ratio microchannel geometries in ambient atmosphere. This is

problematic as open-channel systems carry unique advantages to that of enclosed systems[25]. Namely, their lack of component assembly promotes large-scale manufacturability and channel dimensions that can be scaled below sub-10 μm—toward single particle, for optimal integration with resonant nanostructures. Furthermore, the open access to the full channel enables location-specific surface functionalization, direct sample probing, and the release of unwanted microbubbles that can obstruct enclosed systems[26]. Additionally, without external channel material obstructing the channel viewport, large numerical-aperture microscopy can be used for high-resolution, real-time imaging and reduced optical loss. The latter is specifically important for biosensing via infrared (IR)-absorption spectroscopy in which IR protein resonances compete with the strong absorption of channel material (e.g., polydimethylsiloxane or PDMS) and the ambient saline solution[27–29].

Predominantly, open-channel systems are passively operated via capillary action and thus the scope of fluid processing is limited. The timing and placement of liquid media depends on the skill of the end user and every fluid operation must be predefined during fabrication. This limits versatility and can result in increasingly large spatial footprints on the microfluidic platform (e.g., multiple, large mixing structures between each reaction

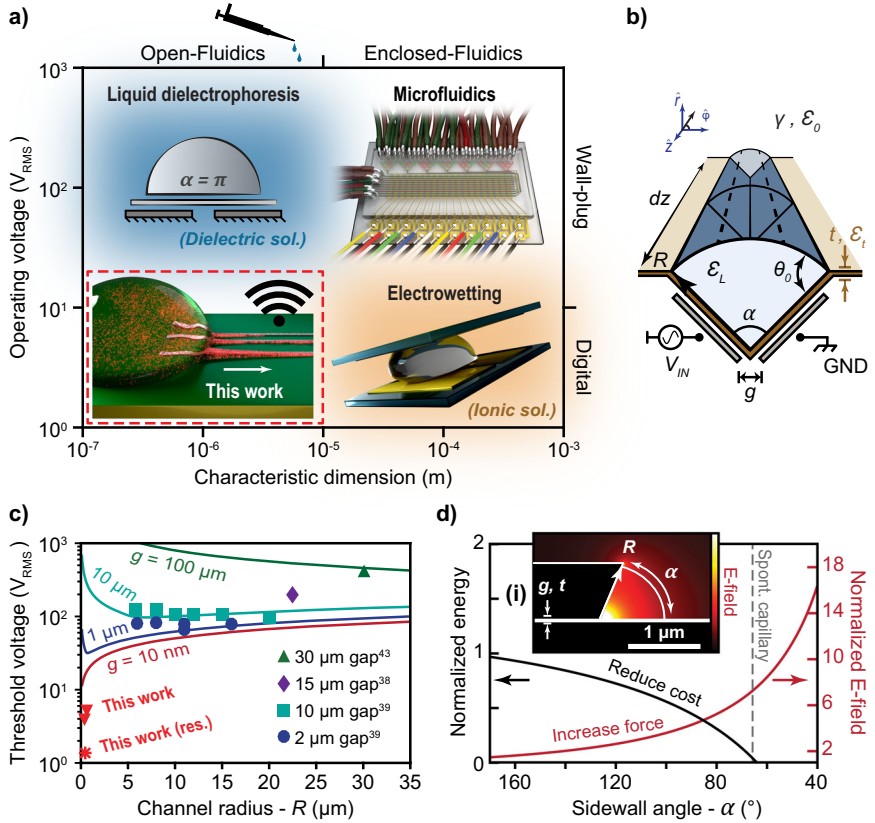

**Fig. 1 Current limitations and a new approach for open-channel electrofluidics. a** Schematic of common fluidic microsystems, comparing operating voltage and characteristic fluidic dimensions. This work offers digital fluidic actuation with micron-scale channel confinement and user-friendly, open-drop loading for either ionic or dielectric solutions. **b** Schematic of an open-micro-electrofluidic (OMEF) channel that travels a distance, $dz$, with a circular-sector cross section of radius $R$ and sector angle $\alpha$. Two electrodes comprise the legs of the sector separated by a gap, $g$, beneath a passivation layer with thickness $t$, dielectric permittivity $\varepsilon_t$, and contact angle $\theta_O$. The liquid channel has a dielectric permittivity of $\varepsilon_L$, surrounded by an environment with a permittivity of $\varepsilon_O$ and surface tension, $\gamma$. **c** Theoretical threshold voltage necessary to pull a channel of deionized (DI) water using coplanar electrodes ($\alpha = \pi$) for various $g$ values as a function of $R$. Literature values are plotted and in good agreement with our work[38,39,43]. A few of our results using a nonplanar geometry are plotted for comparison. **d** Reducing $\alpha$ simultaneously reduces the surface-energy cost to extrude OMEF channels (black curve) until spontaneous capillary action occurs and it also confines E-fields more efficiently within the channel volume (red curve, normalized to the coplanar geometry with the same radius). The inset shows simulation of the E-field distribution for a nanogap-stacked electrode whose sidewall forms the OMEF circular-sector channel. Symbols match the definitions of Fig. 1b. Source data are provided as a Source Data file for Fig. 1c, d.

step)[30,31]. Rather, active control can promote more complicated fluidic handling[25] with smaller footprints[30,31], in which open-micro-electrofluidic (OMEF) technology presented herein offers both confined open channels with active liquid actuation.

For OMEF, a liquid sample is directly placed as a droplet on a planar chip—fabricated using industry-standard microfabrication techniques, followed by the extrusion of narrowly, confined (e.g., sub-10 μm) open-fluidic channels using electric fields (E-fields) (Fig. 1A). Optically driven microfluidics is one example of such a microfluidic system[32]. However, it lacks the scalability of integrated-circuit elements to generate the actuating E-fields with wafer scale, lithographic precision. Liquid dielectrophoresis (LDEP) was developed to utilize these advantages and was able to confine OMEF channels using a radio-frequency (RF)-driving signal applied across closely spaced (~2–10 μm) coplanar electrodes. However, the impact of ST necessitated high operating voltages (i.e., ~50–200 $V_{RMS}$), which became even larger using conductive/physiological solutions[33–39].

This has left a technological void in terms of low-power, active control over open-channel electrofluidics (Fig. 1A). To address this need, we have utilized a surface-energy perspective to understand the limitation of ST on OMEF and motivate the technological development within this microfluidic subspace. By using simple geometric strategies for the electrode configuration (Fig. 1B), we have mitigated these limitations to enable OMEF actuation below 5 $V_{RMS}$ for both dielectric and ionic liquid media (as low as 0.5 $V_{RMS}$) (Fig. 1C). We highlight several on-chip fluidic applications, including solution mixing, lateral-flow protein labeling, particle filtration, and confined transport with IR-absorption spectroscopy of viral capsids. Additionally, resonant circuitry was integrated to reduce the operating voltage below digital levels for promoting portable, point-of-care applications. We showcase this by powering the fluidic device using a wireless RF signal and a near-field communication (NFC) signal emitted from a smartphone. By addressing surface energy and optimal RF focusing, we can tap into Moore's law for equivalent fluidic scaling of active, open-channel systems (Fig. 1a).

## Results

The limiting effects of ST must not be ignored in open-channel platforms as the dimensions of fluidic cross sections are reduced. The energy cost to generate large interfaces with an ambient hydrophobic environment (e.g., air or oil), can be appreciated using a differential Gibb's free-energy description[40,41]:

$$dG = \sum_i \gamma_i \, dA_i - p \, dV - S \, dT \leq 0 \qquad (1)$$

Here, $dG$ is the differential change in Gibb's free energy, which is zero at equilibrium and negative during energy minimization. The surface tension, $\gamma_i$, between a change in surface area, $dA_i$, is summed across all liquid interfaces and is balanced by the pressure difference, $p$, across the liquid–air-phase boundary and the corresponding volume change, $dV$. The system's entropy, $S$, in response to differential temperature changes, $dT$, is ignored assuming constant temperature during initial actuation. The change in surface area and volume of the reservoir drop will be negligibly small during channel formation, thus Eq. 1 can be strictly applied to the microchannel geometry (Fig. 1B).

The Korteweg–Helmholtz relation describes a local pressure difference that can form near a tangential component of an E-field across the surface of a dielectric (see Supplementary Materials)[42]. If this pressure term is inserted into Eq. 1, a condition for the critical electric field necessary to oppose ST for an arbitrary channel geometry is derived (see Supplementary Materials). For a planarized open-channel chip, a circular-sector fluidic cross section is defined with radius $R$ and sector angle $\alpha$

(Fig. 1B). The E-field condition for actuation becomes

$$E_t^2 \geq \frac{4\gamma(\alpha - 2\cos(\theta))}{\alpha R(\varepsilon_L - \varepsilon_0)} \qquad (2)$$

where $\theta$ is the Young's static contact angle formed between the liquid and the electrodes (Fig. 1B), and $\gamma$ is the surface tension of the liquid with the environment (air). The strength of the E-field generated across two electrodes can be increased without raising the operating voltage by instead reducing the gap distance, $g$, between them. We simulated coplanar LDEP electrodes ($\alpha = \pi$) with gap values found in the literature[38,39,43] to predict the effects of ST on the threshold voltage, and plotted these as a function of the channel dimensions (Fig. 1C). The effects of ST as determined using Eq. 2 were in close agreement with the literature for the LDEP coplanar configuration after normalizing the substrate contact angles (see Supplementary Materials) (Fig. 1C).

Interestingly, the nanometer-gap regime confines RF E-fields (nano-RF) with gains faster than the opposing effect of ST (Fig. 1C). However, this necessitates threshold voltages still too large for most nanogaps. Dielectric breakdown[44], Joule heating, and hydrolysis are several concerns in which the latter would require a passivation layer that would likely consume much of the confined E-fields. Equation 2 also assumes a purely dielectric body; for more conductive (e.g., physiological) solutions, suspended charge particles can screen the E-fields and diminish the dielectrophoretic pressure. For a classic LDEP geometry ($\alpha = \pi$), Eq. 2 cannot hold when $E_t \approx 0$ due to geometric constraints. While the frequency of operation can be increased beyond the charge relaxation time for dielectric actuation, for physiologically relevant solutions with high conductivity (~1 S/m), this is usually too high (~100 MHz)[45] to be feasible at 50–100 $V_{RMS}$ operating voltages.

However, Eq. 2 suggests that open-microfluidic channels could exist when $E_t \approx 0$, if a non-coplanar electrode geometry ($\alpha \neq \pi$) is used. This condition can be solved for by setting the dielectrophoretic pressure to zero:

$$\cos(\theta) \geq \frac{\alpha}{2} \rightarrow \theta \leq \sim \frac{1}{2}(\pi - \alpha) \qquad (3)$$

Under small-angle approximations, Eq. 3 simplifies to the Concuss–Finn (CF) limit—a criterion for spontaneous capillary action along the corners of microfluidic channels[40,46]. The contact angle, $\theta$, can be controlled to determine when Eq. 3 is satisfied using the Lippmann equation standard to EWOD applications. As the dielectric pressure term (Eq. 2) is diminished with increasing solution conductivity, the effects of surface wettability become more prominent, and thus OMEF can still be tractable. A threshold voltage for OMEF actuation of conductive solutions can be solved (see Supplementary Materials) when using the same circular-sector cross section defined above (Fig. 1B):

$$V^2 \geq \frac{\gamma t(\alpha - 2\cos(\theta_0))}{\varepsilon_t} \qquad (4)$$

Here, the initial Young's contact angle, $\theta_0$, corresponds to a dielectric passivation layer that covers the electrodes (similar to EWOD) with thickness $t$ and dielectric permittivity $\varepsilon_t$.

Vertically stacked electrodes separated by a nanometer dielectric layer (20 nm $Al_2O_3$) offer a simple electrode geometry favorable to both operating conditions (Eqs. 2 and 4), where the edge of the top electrode defines one leg of the channel sector and the nanolayer defines both the gap separation ($g$) and bottom passivation-layer thickness ($t$) (Fig. 1D). Using finite-element modeling, we observed twofold advantages of reducing the sidewall angle, $\alpha$. The surface-energy cost to produce the OMEF channel is reduced as the area exposed to "hydrophobic" air is decreased, while simultaneously

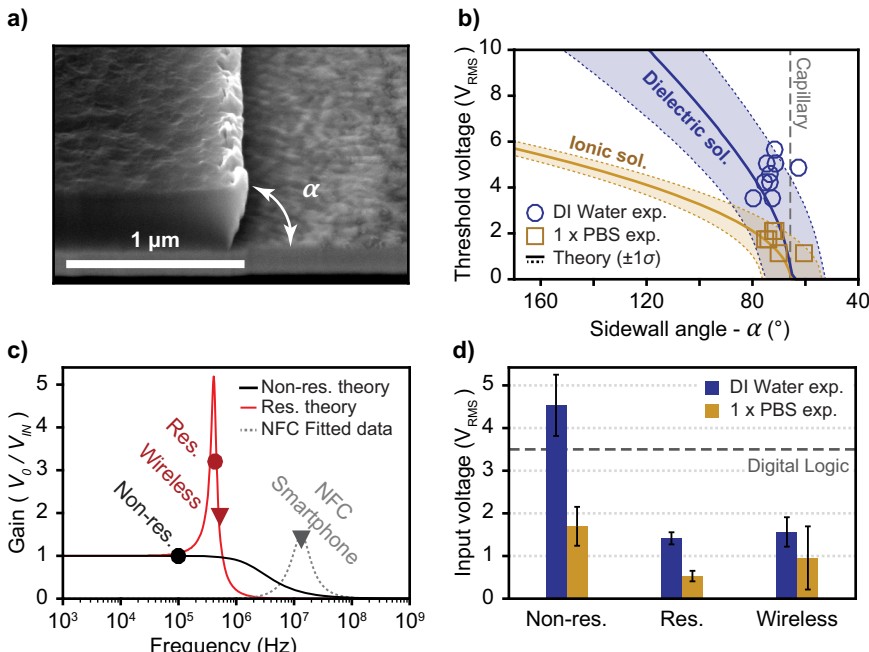

**Fig. 2 Efficient electrode geometry for digitally driven, wireless open-channel fluidics. a** Scanning electron microscopy (SEM) cross-section of our electrode stack after fabrication with a sidewall angle, $\alpha$, clearly defined. **b** Theoretical threshold-voltage curves were plotted as a function of $\alpha$, using our measured device parameters with ±1 standard deviation. This was simulated for both dielectric (DI water) and ionic (PBS) solutions. Experimental data for both regimes are in good agreement. **c** The theoretical and observed voltage gain is plotted as a function of frequency for nonresonant, resonant, and wireless resonant operation. A wireless NFC circuit was also designed and powered using a smartphone to actuate PBS buffer. A theoretical curve to simulate the complex circuitry within the smartphone device was fitted to the observed data using our known circuit elements. **d** The required input voltage experimentally found for DI water and PBS buffer for three different device configurations: wired nonresonant, wired resonant (via a series inductor), and resonant wireless using inductively coupled coils (1–1.5 cm separation). With resonant operation, the input voltage for both dielectric and conductive solutions is below the threshold for a digital logic signal. Error bars are ±1 standard deviation. Source data are provided as a Source Data file for Fig. 1b, d.

increasing the E-field confinement to pull the fluid (Fig. 1D). These factors benefit the physics of both LDEP and EWOD, which can now coincide under a single geometric parameter, $\alpha$ (Eqs. 2 and 4) to actuate both dielectric and conductive liquids, respectively.

## Experimental results and discussion

To test this design, we fabricated stacked nanogap devices (see "Methods"). Figure 2A shows a sample SEM image demonstrating a well-defined sidewall angle $\alpha$. The mean and standard deviation of relevant device parameters were measured (see Supplementary Materials, Table S1–S2) and used to derive the theoretical-threshold voltage for dielectric and conductive solutions to satisfy Eqs. 2 and 4 (Fig. 2B). Experimentally, we tested OMEF actuation using two extremes in liquid conductivity: deionized (DI) water ($4 \times 10^{-4}$ S/m) and high ionic-strength physiological buffer (PBS, 1.52 S/m). Both were fluorescently labeled with Alexa Fluor-594 (AF-594) to visualize the liquid channel and its formation (see Supplementary Materials). Projection lengths varied from 5 to 500 μm with a width of 1–5 μm. We plotted the threshold-voltage results relative to our theoretical curves, and found that they were within one standard deviation of the experimental results from our fabricated device (Fig. 2B). The average threshold voltage to form a microfluidic channel was $4.5 \pm 0.7$ $V_{RMS}$ for DI water and $1.7 \pm 0.5$ $V_{RMS}$ for PBS (Fig. 2D) —a 10-fold and 100-fold reduction relative to the literature LDEP values of 50 $V_{RMS}$ and 200 $V_{RMS}$ for DI water and conductive liquid, respectively.

To push the performance, we placed a low-power inductor in series with the device to create a resonant tank circuit. In this configuration, the input power was more efficiently stored as

E-fields for actuation work due to voltage gains over the device (Fig. 2C). This reduced the necessary input voltage to $1.4 \pm 0.1$ $V_{RMS}$ for DI water and $0.5 \pm 0.1$ $V_{RMS}$ for PBS—both now well below the 5-V amplitude (3.5 $V_{RMS}$) of transistor-to-transistor (TTL) digital logic (Fig. 2D). Furthermore, the inductor was then used as an antenna for wireless power transfer via resonant inductive coupling[47]. Custom-built coil inductors were made to demonstrate wireless OMEF with a 1–1.5-cm separation using input voltages still below TTL thresholds (Fig. 2D).

To demonstrate practical microfluidic applications, we generated arbitrary flow patterns confined to a few micrometers that navigate 90° and 180° turns (Fig. 3A–C and Supplementary Movie 1). In this demonstration, the channel position is precisely known over its entire ~1-mm length from the drop and the averaged flow-rate over five minutes was approximately 1 pL/min (i.e., 0.2% of the drop volume per minute), see Supplementary Materials for more details. This proved appropriate at this scale, as far-less sample volume is wasted in empty tubing and interconnects and the confined channel dimensions promote more rapid analyte interactions, as would occur with classical microfluidics. The capability for upscaling is presented with dense packing of fifty parallel fluidic channels within a microscope view (Fig. 3D), which were used to transport fluorescently labeled norovirus GI.1 virus-like-particles (VLPs) into precisely confined microchannels for Fourier transform-infrared (FTIR) absorption spectroscopy measurements (Fig. 3E and F and Supplementary Movie 2). FTIR spectroscopy is known to experience considerable losses from water absorption in biological samples (e.g., large EWOD droplets) and channel material (e.g., PDMS in classical microfluidics)[27–29]. Therefore, this open-channel fluidic platform is ideal for FTIR biosensing, and was used to detect norovirus

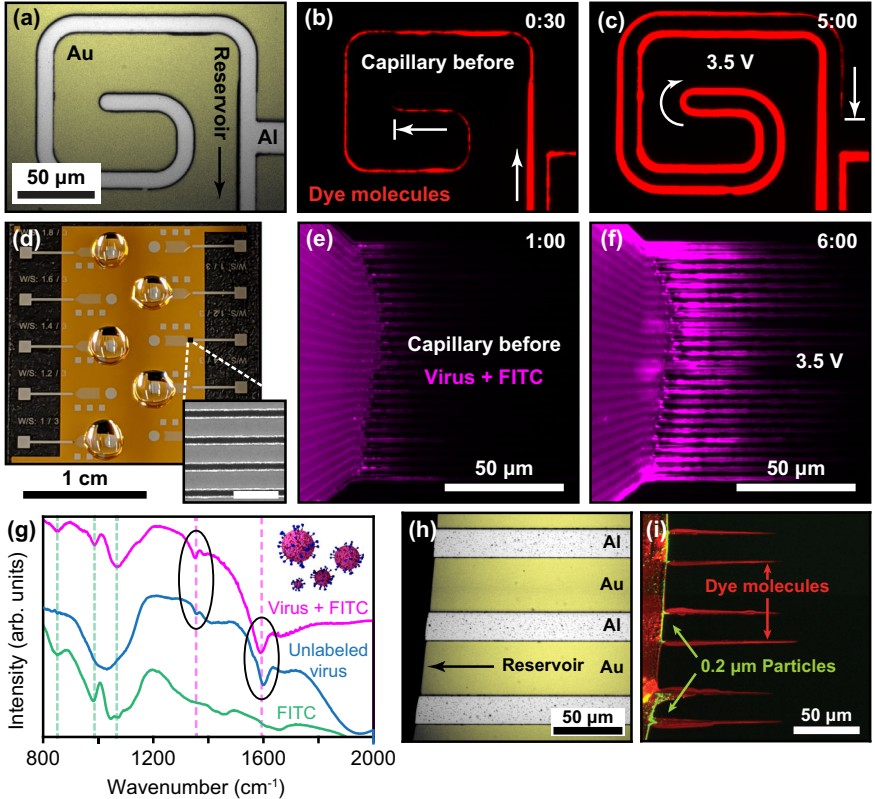

**Fig. 3 Fluidic applications demonstrating high channel-density parallelization and low-loss viral biosensing. a** A spiral-top electrode is patterned out of aluminum, with the bottom gold electrode colored yellow. A drop of PBS is positioned out of view. **b** Using the device shown in (**a**), spontaneous capillary flow is observed using a fluorescent dye due to a sidewall exceeding the capillary limit. The liquid channel travels ~684 µm from the drop. **c** After 3.5 $V_{RMS}$ voltage is applied, the liquid channel wraps around the outer edge of the top electrode and travels an additional ~566 µm in 5 min., for a total length of ~1.25 mm from the drop and 5 µm in width. **d** An image of a sample chip with five droplets placed for actuation. The inset shows an SEM image of a dense line-array used for channel parallelization (scale bar = 5 µm). **e, f** Fluorescent image of parallel fluidic channels used to extrude FITC-labeled norovirus-like particles, before (**e**) and after (**f**) voltage is applied. Images were artificially colored magenta to distinguish from other experiments. **g** Infrared spectroscopy was used to confirm the presence of viral material within the channels compared with fluorescent background. Dips unique to the virus samples are circled in black. Source data are provided as a Source Data file. **h** Three parallel top electrodes with larger separation were used for particle filtration. **i** After applying 3.5 $V_{RMS}$, parallel microchannels formed along the edges of the top electrodes, as confirmed using fluorescent Alexa Fluor-594 molecules (red). Larger 190-nm polystyrene beads (green) spiked into the solution are filtered from entering the microchannels due to low-voltage dielectrophoresis.

VLPs by distinguishing viral protein peaks from a control (Fig. 3G). Future iterations could utilize the same actuating, line-array electrodes as additional FTIR resonators to further boost detection sensitivity[28,29]. We have also demonstrated active filtration of large fluorescent polystyrene (PS) particles (green) from fluorescent AF-594 dye molecules (red) as they are extruded into microchannels (Fig. 3H, I). This is achieved via the strong dielectrophoretic forces that repel PS particles at the channel entrance, and depends on the conductivity of the solution, the size and dielectric permittivity of the particles, the frequency of operation[48], and the microchannel height (see Supplementary Materials). This active filtration greatly improves upon passive capillary where undesired large particles that find their way to the channel entrance would obstruct flow.

We also demonstrated active solution mixing and lateral-flow protein labeling using green fluorescent protein (GFP) and AF-594 dye (Fig. 4A, B and Supplementary Movies 3 and 4). After applying bias, the proteins and dye molecules are pulled to fill the parallel channels and mixed into the opposite reservoirs. Using a fluorescence-intensity metric, the increased GFP fluorescence in the AF-594 reservoir and vice versa could be tracked over time relative to their intensity from their original reservoirs to account for photobleaching (see Methods, Fig. 4B). Within the OMEF channels themselves, conglomerates of GFP were extruded from

the drop and were labeled with AF-594 (seen as red after the AF-594 solution passes through) with locations confirmed using native GFP fluorescence (Fig. 4A). The capability to perform localized chemical labeling, mixing and filtering are important advantages for active microfluidics as compared to droplet-based EWOD, and represents important first steps toward more complex operations[49].

Finally, to demonstrate the possibilities for this platform to integrate with future point-of-care technologies, we have performed wireless, smartphone-driven actuation. By designing a wireless power-transfer circuit at the NFC frequency (see Supplementary Materials, Fig. 2C), we could achieve wireless OMEF actuation of physiological solution (i.e., PBS) using the power from a signal emitted by a smartphone (see Supplementary Materials, Fig. 4C, D and Supplementary Movie 5).

In summary, we sought to fill an important void in fluidic microsystems by addressing the fundamental challenge of ST in active, open-channel designs. Using a combination of nano-RF-field focusing and a lower surface-energy electrode geometry, we demonstrated OMEF actuation using a wireless, digital RF power signal. We highlight the technological potential by showcasing wireless lateral flow for localized protein labeling and the extraction of viral capsids for efficient FTIR biosensing. Free from external tubing, top covers, and wire connects, this method is

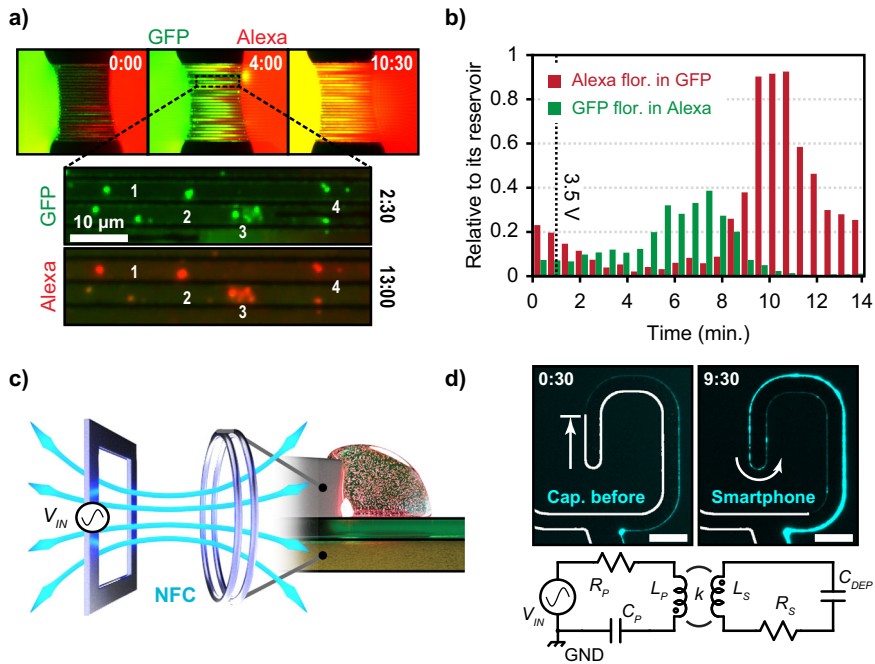

**Fig. 4 Wireless smartphone actuation and lateral flow protein labeling. a** A green fluorescent protein (GFP) reservoir drop (green) and Alexa Fluor-594 reservoir drop (red) were mixed using 25 parallel channels after applying 3.5 $V_{RMS}$. Conglomerates of GFP were extruded from the GFP reservoir into individual OMEF channels and were labeled red by the Alexa dye after mixing using amine chemistry. **b** Alexa fluorescence intensity in the GFP reservoir is plotted versus time (red) and normalized to its source-reservoir fluorescence at the same time point. Similarly, GFP fluorescence inside the Alexa reservoir is also plotted versus time (green). An exchange in reservoir material is observed for both solutions. Source data are provided as a Source Data file. **c** Schematic demonstrating wireless inductive coupling of magnetic fields to actuate liquid. **d** Demonstration of wireless smartphone actuation of PBS using an NFC signal. Images are from before (left) and after (right) smartphone-powered actuation was applied, where the solution was pulled an additional ~200 μm beyond the capillary action while increasing the channel width throughout its entire length. The electrode edge is outlined in white for reference. Scale bar is 25 μm. Below is an example of a resonant wireless power-transfer circuit using a primary inductor ($L_P$), capacitor ($C_P$), and resistor ($R_P$) (LCR) circuit to transfer power to a secondary LCR with an inductance ($L_S$), resistance ($R_S$), and circuit capacitance of our device, $C_{DEP}$.

especially ideal for real-time, large numerical-aperture imaging of channel progression and is ideal for infrared biosensing techniques (e.g., FTIR) that suffer from top-cover and water absorption losses. Our strategy simplifies fluidic scaling of channel dimensions to lithographically defined electrodes with no post-assembly or channel-curing steps required. The ability to wirelessly drive open fluidic channels—confined to the same scale as the analytes, using a common smartphone, unveils great possibility within the realm of open microfluidic handling, vastly simplifying operation and thus serving a broader and more diverse community of users.

## Methods

This study aims to design a device capable of pulling confined, open channels of liquid (≥0.5 μm in width) using localized electric fields without pumps or tubing. Further, we aim to push operating conditions below 5 $V_{RMS}$ (within the range of transistor logic), which is below standard EWOD and LDEP operation for both dielectric and conductive liquids, and demonstrate practical on-chip, microfluidic functions. Finally, we aim to use resonant inductive coupling to achieve wireless actuation of both dielectric and conductive liquids. Our prespecified hypothesis was that a reduction in the gap spacing between coplanar electrodes to tens of nanometers would generate intense local electric fields capable of pulling confined liquid channels. This was modified after initial theoretical simulations and preliminary experiments (using a 20 nm $Al_2O_3$ coplanar gap) revealed that dielectric breakdown occurs before a force large enough to overcome surface tension could be generated. Our hypothesis was subsequently modified as follows: a circular-sector channel geometry with a sector angle less than 180° and a nanogap-spaced pair of electrodes is able to break surface tension (preventing dielectric breakdown) and pull open-liquid channels with 5 $V_{RMS}$ or less for both dielectric and conductive solutions. The specific methodology for designing, characterizing, measuring, and processing data is included below.

**Statistics**. The threshold voltage was tested repeatedly for both deionized (DI) water and 1 × phosphate-buffered saline (PBS) over a total of 32 devices. For DI

water-threshold experiments, 18 devices were used (ten wired and eight wireless). Two outliers were found (one wired and one wireless), which could not pull liquid before the drop evaporated with operating voltages >6 $V_{RMS}$, and were omitted from this study. This lack of actuation could be attributed to defects or features at the microchannel entrance that prohibited channel formation (see "electrode edge characterization" in the Supplementary Materials). Four wired connections were used to simultaneously observe resonant threshold conditions (see "Nonresonant and resonant data acquisition" section below). Using a two-tail, t-distribution with a defined 95% confidence interval (p-score = 0.05), the error found in the reported mean threshold values for all DI water-actuation experiments was 0.6 $V_{RMS}$ or lower of the true mean using the corresponding sample sizes (see Table S3).

For PBS-threshold experiments, 14 devices were used (six for wired connection and six for wireless) with two outliers during wireless actuation in which inefficient power was transferred. This is attributed to a nonresonant device capacitance where it was observed that wireless PBS experiments were more sensitive to deviations in the device capacitance, see "Wireless data acquisition" section below. Four of the wired connections were used to simultaneously observe wired, resonant threshold conditions (see "nonresonant and resonant data acquisition" section below). Using a two-tailed t-distribution and a defined 95% confidence interval (p-score = 0.05), the error in the mean-threshold values for wired PBS actuation (resonant and nonresonant) was 0.5 $V_{RMS}$ or less of the true population mean, it was 0.8 $V_{RMS}$ or less of the true population mean for wireless PBS actuation using their corresponding sample sizes (see Table S3).

All subsequent mean values with reported bounds are one-sample standard deviation, with the inclusion of the number $n$ of measurements made. In addition to the 32 standard devices, chips used for spiral-flow patterns, particle separation, protein labeling, viral extraction, and smartphone actuation were made for demonstrating proof-of-concept applications with the experimental conditions outlined below. Before experimentation, devices were characterized and only those within the target-fabrication tolerance were considered for testing. This included characterization of the edge profile, oxide gap width, substrate contact angle, and electrical properties. Details on these characterization steps and results can be found in the Supplementary Materials. Additionally, during experiments, the voltage dropped over the device was continually monitored in a parallel configuration using an oscilloscope (internal resistance of 1 MΩ, Tetronix) to confirm proper transfer of voltage to the fluidic device.

**Fabrication of stacked electrodes**. Fabrication starts with a 500-μm-thick Borofloat 33 glass wafer (University Wafer) that is cleaned in a standard piranha solution and thoroughly rinsed with deionized (DI) water. Rectangular bottom electrodes (15 × 23 mm) were then patterned using standard photolithography and electron-beam-evaporated gold deposition (50 nm Au, with a 3 nm Cr-adhesion layer) (Fig. S1A—Step 1). The wafer was then cut into individual 25 × 25 mm chips and thoroughly cleaned (1165 Microposit Remover, 60 °C sonication bath). A 20 nm layer of $Al_2O_3$ was grown using atomic-layer deposition (ALD) at 250 °C as 20 nm provided a higher device yield as compared with thinner layers. The thickness of this layer defines the gap spacing, $g$, between the bottom and top electrode and passivation-layer thickness, $t$ (Fig. S1A—Step 2). This is followed by a final photolithography step to pattern the top electrodes, contact pads, and a placement pad to guide the eye when loading the droplet. A 400-nm layer of electron beam-evaporated Al was then grown (Fig. S1A—Step 4), followed by lift-off. Al was chosen as a low-stress, soft metal capable of maintaining a 400-nm thin film without flaking (compared with Cu) and is relatively cheap. The top-electrode dimensions used for threshold voltage experiments were 50 μm wide and 10 mm long. Dimensions for viral extraction and protein mixing consisted of 25 parallel lines (resulting in 50 fluidic channels per edge) with a 3-μm pitch, that were 1 μm wide and 150 μm long (Fig. 3D). Dimensions for particle filtration consisted of three parallel lines (40 μm pitch) that were 20 μm wide and 10 mm long (Fig. 3H). To control the sidewall angle using lift-off, the thickness and type of photoresist (PR) is important. To create an electrode edge with an acute sidewall angle, $\alpha$, a positive PR should be used with a thickness of at least 3× the desired height of the top electrode, $R$, to facilitate a clean lift-off. We used a ~2-μm-thick PR layer of AZ1518 (MicroChemicals) for a 400-nm-tall top electrode (Fig. S1A—Step 3). The average $\alpha$ was ~72° (Fig. S1A—Step 5) (see Supplementary Materials), which satisfies the Concuss–Finn limit when the substrate contact angle with the liquid reservoir is less than ~50° (Eq. 3). Titania ($TiO_2$) is known for being a biocompatible and stable passivation layer[50] with a relatively high-K dielectric constant of 80–100[51], which is ideal for reducing its effective oxide thickness and making its presence electrically negligible. Therefore, a final 5 nm of $TiO_2$ was PEALD-grown over the complete device at 250 °C to passivate the electrodes and establish roughly the same contact angle across all materials of the device. The 20-nm $Al_2O_3$ and 5-nm $TiO_2$ films were measured using single-wavelength ellipsometry (Gaertner Scientific Corporation) on a silicon test piece included with deposition. An image of a complete chip used for threshold-voltage experiments is provided in Fig. S1B, and the drop-placement pad can be seen with the naked eye (Fig. S1C). Dielectric breakdown was measured at ~12 V using DC measurement (Fig. S2).

**Preparation of sample solutions**. Sample solutions were created using DI water and 1 × PBS (pH 7.4, Sigma-Aldrich), with measured conductivity values of $4 × 10^{-4}$ S/m and 1.52 S/m, respectively, as measured with a B-771 LAQUAtwin (Horiba Scientific). To visualize the channel during its formation, fluorescent dye molecules were mixed into both buffer solutions using 5 μM Alexa Fluor-594 (AF-594) carboxylic acid ($\lambda_{ex}$: 590 nm, $\lambda_{em}$: 617 nm, molecular weight: 819.85 Da, Thermo Fisher Scientific). Protein mixing and labeling experiments utilized green fluorescent protein (GFP) modified with a polyhistidine tag ($\lambda_{ex}$: 487 nm, $\lambda_{em}$: 508 nm, GFPSpark, Sino Biological). This was mixed at a concentration of 4.5 μM in 1 × PBS buffer solution for sample mixing and amine binding with the AF-594. For virus extraction, Norovirus GI.1 virus-like particles (VLPs) (20–40 nm diameter; The Native Antigen Company) were fluorescently labeled for imaging using a commercial FITC labeling kit ($\lambda_{ex}$: 495 nm, $\lambda_{em}$: 519 nm, Abcam). This solution was then diluted with PBS, such that the virus mass concentration was 7.4 μg/mL. Samples containing unlabeled VLPs and FITC alone at the same dilution in PBS were made for control measurements. For experiments demonstrating filtration, solutions were made with fluorescent 190-nm-diameter polystyrene (PS) beads ($\lambda_{ex}$: 470 nm, $\lambda_{em}$: 525 nm, Surf Green, Bangs Labs). The PS bead stock solution was diluted and mixed with the AF-594 solution to a final concentration of 5 μM AF-594 dye with 100 μg/mL PS beads in both DI water and 1 × PBS buffer.

**Preparation of loop antennas**. For demonstrating wireless actuation, two different homespun solenoid-inductive antennas were made using 24-gauge silver plated copper wire (OD: 0.031 inches, Lapp Tannehill) to inductively couple power. They were characterized using an LCR meter (4284 A Precision LCR Meter, Hewlett Packard) and a caliper (AOS Absolute Caliper, Mitutoyo). The primary coil used for transmitting power had a measured inductance of 10.1 μH (number of turns = 10, diameter = 3.6 cm, height = 0.3 cm, and quality factor = 25). The secondary coil used for collecting power was electrically connected to our device and had a measured inductance of 51.9 μH (number of turns = 30, diameter = 3.5 cm, height = 0.6 cm, and quality factor = 50).

**Nonresonant and resonant data acquisition**. Channel formation was observed using an upright microscope with an extra-long working distance 50× air objective (NA 0.55, Nikon). A 2–5 μL droplet of sample solution was placed on a circular drop placement pad (to guide the eye), so that it overlapped with the top electrodes. A sinusoidal AC signal was then applied across the nanogap using a function generator (Hewlett–Packard) with 50 Ω internal resistance (Fig. S4A). To visualize

channel formation, a white light laser-driven light source (LDLS, Energetiq) was used, with a red filter cube ($\lambda_{ex}$: 562 nm, dichroic: 594 nm, $\lambda_{em}$: 593 nm; Semrock) for imaging the AF-594 dye and general channel evolution and a green filter cube ($\lambda_{ex}$: 470 nm, dichroic: 495 nm, $\lambda_{em}$: 525 nm, Chroma) for imaging target analytes (i.e., GFP, FITC-VLPs, and PS particles). The evolution of the channel was recorded in 1 s intervals (2 × 2 pixel binning, 400 ms exposure, Micro-Manager) using a charge-coupled device (CCD) camera (CoolSNAP $HQ^2$, Photometrics). With our 50× objective and pixel-binning settings, the image pixel size corresponded to 260 nm, which meets the Nyquist sampling criterion to resolve the theoretical diffraction limit for the emission maximum of the AF-594 molecules (617 nm/(2 × NA) ≈ 561 nm). Video sequences were recorded as 16-bit TIF image stacks in which individual frames were pulled to form text figures. The corresponding video files (converted to AVI format) with their appropriate timestamps are included as Supplementary materials. Images directly compared in the text maintained identical contrast/brightness settings for equal-intensity comparison and linear look-up tables were applied to aid visualization.

During the threshold-voltage experiments, the driving signal was set to 100 kHz and was incremented every 60 s by 0.5 V amplitude (0.35 $V_{RMS}$), until a channel was observed with a width of 1 μm or greater and a length greater than 5 μm. This condition was chosen to ensure that actual channel formation could be differentiated from intensity fluctuations at the electrode edge due to light scattering from the reservoir drop. Since the average electrode height was ~850 nm (Table S1), a confirmed microchannel radius should be similar in size, and thus 1 μm or greater was a conservative condition to confirm the presence of actuated liquid. Since the threshold voltage does not depend on the microchannel length, $dz$ (see Supplementary Materials section), lengths greater than 5 μm were subjectively chosen as a liquid deformation, with 5:1 aspect-ratio microchannel dimensions to experimentally determine the first onset of formation.

Resonant-threshold voltage was recorded in like fashion, with a 100 μH inductor wired in series (Fig. S4B) and amplitude increments of 0.1 V (0.07 $V_{RMS}$). Before starting the experiment, the resonant frequency was found after placing the drop and finding the maximum voltage across the device with an oscilloscope (Tektronix) using a small-amplitude driving signal (~100 mV). The gain was defined as the ratio between the voltage drop across the device and the voltage input to the circuit. The average gain was 3.2 ± 0.5 ($n = 8$), using our tank circuit with an average resonant frequency of 432.5 ± 71.1 kHz ($n = 8$). The input voltage and voltage drop across the device were recorded during the liquid-channel formation using an oscilloscope. Post experiment, an SEM cross section was taken of the channel at a region near the channel, and the sidewall angle was measured (see Supplementary Materials). The experimental gains were plotted on top of our theoretical curves (Fig. 2C). When demonstrating arbitrary flow paths, a 3.5 $V_{RMS}$ (100 kHz) driving signal was applied and recorded for 5 min with no inductor. A spiral complex flow pattern of DI solution is included in the Supplementary Materials (Figs. S5A, B and D). Drops lasted ~10 min before fully evaporating.

**Lateral-flow protein labeling**. Protein mixing and labeling utilized the 25 parallel top-electrode design, in which a 4-μl drop containing GFP and another 4-μl drop containing AF-594 were placed at opposite ends of the electrode array. A wired nonresonant circuit (Fig. S4A and Supplementary Movie 3) (100 kHz) and wireless resonant circuit (Fig. S4C and Supplementary Movie 4) (200–300 kHz, 1.2 cm coil separation) were used with an input voltage of 4.2 $V_{RMS}$ and 3.5 $V_{RMS}$ driving signal for actuation, respectively. Images were acquired every 30 s with both filter cubes to visualize the progression of GFP and AF-594 fluorescence during mixing, and composite images were prepared (Fig. 4A). After both drops dried (~15 min), conglomerates of GFP within the channels can be seen labeled by the AF-594 dye. This dye contains a carboxylic acid group for binding to the histidine tag of the GFP protein (or its other 36 free amine groups). This experiment was repeated over three different devices, in which mixing and labeling of GFP clumps were observed. The intensity was averaged over the portion of the drop within the field-of-view to observe the relative increase in AF-594 fluorescence within the GFP reservoir and vice versa (Fig. 4B). These values were normalized to their source reservoir to account for photobleaching effects.

**Virus-particle sensing via infrared-absorption spectroscopy**. Viral extraction was recorded under fluorescent microscopy using solutions containing VLPs, FITC-labeled VLPs, and FITC solution (using the same camera settings outlined above) for five minutes (Fig. 3E–G and Supplementary Movie 2). The 25 parallel top electrodes (50 fluidic channels per edge) (Fig. 3D) were used with a wired nonresonant circuit (Fig. S4A) and a 4.2 $V_{RMS}$ (100 kHz) driving signal for actuation. The evolution of the channels was recorded every second for five minutes. Once dried, infrared-absorption spectroscopy was performed on a sample area of 100 × 100 μm² over the channel region. A Thermo Fisher Scientific Nicolet iS 50 FTIR spectrometer equipped with a liquid-nitrogen-cooled mercury–cadmium–telluride detector was employed for measuring in the reflection mode. Reflected signals from the sample area were collected through the 15 × Reflechromat objective (0.58 N.A.) under acquisition settings of 200 scans with 4 cm⁻¹ resolution. This was measured over three different devices in which one contained VLP capsids, one contained background FITC-label, and one of FITC labeled VLP capsids. Absorption dips exclusive to solutions containing the

VLP capsids could be differentiated from peaks exclusive to solutions with FITC, confirming the presence of viral material within the channels (Fig. 3G).

When demonstrating active particle filtration, an image was taken before applying voltage using both filter cubes. A 3.5 $V_{RMS}$ (100 kHz) driving signal was then applied for 5 min before recording another image using both filter cubes for comparison. In order to capture all six channels with a limited microscope field-of-view, two images (with each filter cube) were taken and merged to observe all six channels in a single frame and create Fig. 3H, I and S5C. With PBS, negative dielectrophoresis (DEP) repelled particles from entering the microchannel, allowing the channels to form while rejecting the PS beads (Fig. 3I). However, for DI, positive DEP trapped the PS beads at the channel entrance and subsequently impeded channel formation. This was retested at a higher operating voltage (5.3 $V_{RMS}$) and channel formation was still impeded (Fig. S5C). Further discussion on DEP filtering is provided in the Supplementary Materials.

**Wireless data acquisition**. Wireless data was collected using a similar methodology to that outlined above. The primary coil ($L_p = 10 \, \mu H$) was connected to a function generator, and the secondary coil ($L_s = 50 \, \mu H$) was connected to our device. Air separation between the coils was fixed at 1.2 cm throughout the entirety of the experiment (Fig. S4C). After placing the drop on the device, a small-amplitude driving signal (~100 mV) was used to find the resonant operating frequency of the device (i.e., the frequency that produced the largest voltage gain across the device, as measured by an oscilloscope). The input voltage was then increased, until a microchannel formed as defined above, and the gain and resonant frequency were recorded. We found experimentally that wireless PBS-threshold experiments were more sensitive to variations in the wet-device capacitance, resulting in less predictable operating resonant frequency. To mitigate this, a 1-nF capacitor was placed in parallel, in which case the same threshold voltage was still dropped across both the device and added capacitor, which made operation more robust to fluctuations in the wet-device capacitance (Fig. S4C). This was done solely for wireless PBS measurements, and the specific capacitance value was chosen to keep the total capacitance similar to the previously measured averaged wet-device capacitance of 1.5 nF (Table S4). This circuit configuration was implemented as proof of concept for wireless actuation of liquid, with further optimization reserved for future work.

For NFC actuation, a Google Pixel 3a smartphone with an NFC application running (Class NfcF JIS 6319-4, NFC Tools, wakdev) was brought in close proximity (<1 cm) to the 10 μH coil (Fig. S4D) for 5 min, while the actuation of PBS buffer was recorded using a microscope and CCD (Supplementary Movie 5). The voltage and frequency drop across our device was monitored in real time using an oscilloscope to confirm wireless transfer of NFC power. The voltage ranged between 1 and 10 V amplitude, depending on how the phone was held (i.e., angle, small shifts in height) with an average value of 4.8 $V_{RMS}$. Per NFC standards, the driving frequency was 13.5 MHz. This experiment was performed three times with PBS, and two of the three repeats yielded discernable liquid actuation (Fig. 4D). One experiment was attempted using DI water, in which no actuation was observed. Since the average voltage was near the threshold criteria for DI water, we believe that actuation was not observed due to the difficulty in maintaining consistent voltage above threshold. Having demonstrated this as proof of concept, more optimal resonant circuitry can be developed in future work to yield better transfer of power for wireless smartphone actuation of DI water.

## Data availability

Most data generated or analyzed during this study are included in the published article or Supplementary Material. Source data are provided with this paper for Figs. 1C–D, 2B–D, 3G, 4B, S2, S7, and S8, including relevant MATLAB scripts. Due to file-size limits, the raw and processed TIF image-sequence files of Figs. 3A–C, 3E, F, 4A and D can be provided upon reasonable request. Source data are provided with this paper.

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

## Acknowledgements

This research was supported by the National Science Foundation (NSF ECCS 1610333). C.T.E. and D.J.K. acknowledge support from the NSF Graduate Research Fellowship Program. S.-H.O. further acknowledges support from the Sanford P. Bordeau Endowed Chair at the University of Minnesota and the McKnight Foundation. Device fabrication was performed in the Minnesota Nano Center at the University of Minnesota, which is supported by the NSF through the National Nanotechnology Coordinated Infrastructure (NNCI) under Award Number ECCS-1542202. Parts of this work were carried out in the Characterization Facility, University of Minnesota, which receives partial support from the NSF through the MRSEC program.

## Author contributions

S.-H.O. conceived the general concept, C.T.E developed the theory to include surface tension and utilize geometrical gains, ran simulations, designed and fabricated the device, designed circuits, and ran experimentation and data analysis. D.Y. aided fabrication and ran and interpreted infrared-absorption measurements. D.J.K and P.R.C milled and imaged the electrode-edge cross sections for sidewall characterization. All authors contributed in preparing the paper.

## Competing interests

The authors declare the following competing interests: an international patent with inventors C. T. Ertsgaard, S.-H. Oh, 2020. PCT Application PCT/US20/61553, filed November 20th, 2020. Patent pending. Patent covers specific design parameters outlined within this paper. All other authors declare no competing interests.
