## [Peer Review File · Nature Communications]

Reviewers' comments:

Reviewer #1 (Remarks to the Author):

Congratulations on reporting a very good proof-of-concept work for low voltage wireless actuation of open microfluidics. I quite enjoyed reading it, to be really honest. Well done!

Reviewer #2 (Remarks to the Author):

The authors developed a portable electrically-powered open microfluidic device that can be wirelessly controlled via NFC. The authors demonstrated the capability of this device to flow in channels with 90 degree turns, conduct particle filtration, particle transport and detection, and solution mixing. While the authors have presented interesting results in this work, I recommend that the paper undergoes major revision.

For the revision, I suggest the authors provide additional context behind the motivation of this work. The authors can elaborate on the significant advantage of their device compared to capillary-driven open microfluidics (see "Suspended microfluidics" published by Casavant et al., PNAS (2013)). From Figure 1D, Figure 3, and Figure 4, there is a clear advantage of this device over capillary-driven microfluidics, so I think authors should definitely make an effort to elaborate on these advantages.

From established literature, I would like to direct the authors to the paper, "Electro-actuated valves and self-vented channels enable programmable flow control and monitoring in capillary-driven microfluidics" (Arango et. al., Science Advances (2020)). In this paper, the closed microfluidic device developed by Arango et. al. seems to be comparable to this work (<5V to actuate fluid flow, compact and portable, and fluid control with electric fields). Arango et. al. also claims flow rate significantly higher than flow rate reported for this device in this work (0.5 – 3 nL/s vs 1 μ L/min). The difference between this work and work from Arango et. al. is that the channel is open in this work while device from Arango et. al. is closed. Are there any significant advantage/s of open microfluidics in this device? Please elaborate in writing and I think an application that demonstrates the advantage/s of the openness of this device would strengthen this work.

Overall, I find that it was difficult to piece together the significance of this device. For example, it is not clear why this device has to be an open microfluidic device when closed microfluidic devices can

perform all these tasks better (see previous paragraph on the work by Arango et. al.). Also, how does this device advance the field of open microfluidics? I suggest conducting major revision to the introduction and conclusion.

Lastly, here are comments for the authors.

Comments on overall work

1. What is the advantage of this work compared to capillary-driven open microfluidics?
2. Does flow speed change with voltage applied?
3. Is there significant Joule heating that lead to unwanted droplet evaporation?
4. Would the channel material affect the flow speed of the device?
5. In Supplementary Information, please show appropriate steps for the derivation of Equation S4.
6. Regarding the calculation of flow rate, did the authors verify if the liquid fully fills the channel cross section when fluid flow is actuated?
7. Can this device operate when submerged in another liquid (i.e. replacing air with another liquid)? If yes, is there a liquid combination that lowers threshold voltage for OMEF actuation and increases flow rate?
8. Present an application that showcase the advantage/s of the openness of this device.

Comments on figures

1. Figure 1A is inaccurate as “Microfluidics” has characteristic dimensions typically ranging from μm to mm.
2. Please include label for each part of the device in Figure 1B.
3. Please include an intermediate frame for Figure 3B-C, Figure 3E-F, and Figure 4D as it is hard to tell what is happening with the current frames presented.

Reviewer #3 (Remarks to the Author):

This paper represents an innovative design of open fluidic microsystem that addresses the fundamental challenge of surface tension in open-channel design. Using a combination of nano-RF field focusing and a lower surface energy electrode geometry, they demonstrated OMEF actuation using a wireless, digital RF power signal. They show the actuation of microchannels as narrow as $1\ \mu\text{m}$ using voltages as low as $0.5\ \text{VRMS}$ for both dielectric and physiological liquids. They further demonstrated the practical fluidic applications including open mixing, lateral-flow protein labeling, filtration, and viral transport for efficient biosensing. This system is coupled with resonant wireless

power transfer for remote actuation, with smartphone-driven fluidics presented to fully showcase the potential of this open-chip technology.

Though thorough experimental observations are performed, there are several fundamental aspects which the authors did not address properly. I may recommend publishing this work on 'Nature Communication' if the authors can address my following comments and overhaul the manuscript thoroughly in terms of additional experiments and physics.

1 In equation 2, for a planarized open-channel chip, what is the limiting condition of the electrodes spacing for maximum E_t as it doesn't show any direct dependence on 'g'.

In case of vertical stacked electrodes, what is the deciding factor for choosing 'g = 20 nm'?

2. What is the passivation layer of 5 nm signifies here as both the g and t is considered here as same?

3. Is there any dependence of projection length and width on the electrode geometry (like a and R)?

4. How does the energy cost in Fig. 1D increases with a as fabricating a = 0 and p is easier than a $\neq 0$ and p, so it should follow a parabolic trend here instead of the existing one.

5. What is the reason of using a $\sim 71^\circ$ in this experiment, as in Fig. 1D, a close to 90° gives the optimum setting for getting less driving force to drive the liquid and less energy cost?

In connection to this, how does the author used Pos. PR to selectively coat with an obtuse side wall angle to pattern the top electrode in Fig, S1? Please explain it properly.

6. At what frequency the average threshold voltages in Fig. 2B are sufficient to form a microfluidic channel for both DI water and PBS? What is the limiting setting for VRMS and frequency for avoiding Joule heating here?

7. Is there any influence of DEP force on the intensification of voltage gain in Fig. 2C for Resonant circuit at 1 MHz frequency? There is highly likely possibility of having the effect of DEP force on the gain due to the electrode geometry. Please justify the reason of getting a high gain at 105-106 Hz in Fig. 2C.

Reviewer #4 (Remarks to the Author):

The manuscript by Ertsgaard et al. proposes a liquid and particle actuation mechanism in microfluidics using an open-channel architecture and wireless electrical power delivery. The manuscript is well-written, has nice and clear figures, and looks complete with extensive theoretical background, modeling, and experimental results.

Comments:

1- I was not familiar with the term “Open-micro-electrofluidics (OMEF)” and I could not find it on the internet. I think it is a term coined by the authors to also include open-channel EWOD and open-channel DEP. To me, this worked looked like a combination of surface-tension driven liquid pumping (e.g. Gosh et al. Lab Chip, 2014), open-channel capillary systems (e.g. Berthier et al. Anal. Chem. 2019) and digitally controlling the flow of continuous liquids using electrowetting (e.g. virtual electrowetting channels, Banerjee et al. Lab Chip, 2012). But overall, I think it is a novel implementation by its own.

2- Open-channel microfluidics clearly bring some advantages for fabrication (no sealing required) and operation of devices (removing air bubbles, easily merging multiple liquids). But to me the real benefit is to be able to integrate bioreagents to microchannels because it's often the case that sealing/closing step requires high temperature or the use glues or solvents that are not compatible with bioreagents. Unfortunately, the authors do not talk about this advantage and the microfluidic architecture does not look compatible with pre-integration of reagents. Although the final sealing step is omitted in the open-channel architecture, the fabrication process looks still complicated and demanding (e.g. the use of ALD) compared to other open-channel implementations. It was difficult for me to see the motivation behind open-channel architecture for the use-cases demonstrated in the manuscript.

3- Low-power actuation and smartphone compatibility are great features. But it is hard to justify without knowing the target application. If it is for point-of-care diagnostics, not being able to integrate reagents would be a bottleneck I think (my previous comment). If it is more for a research platform, then the smartphone compatibility may not be as important. The manuscript demonstrates some nice use cases, but again it is difficult for me to justify why those applications were chosen and in what context.

4- Although being impressive and convincing, I think the submitted work does not enable a real flow control mechanism for capillary microfluidics. As can be seen in the videos, the liquid advances in the channels by capillarity without applying any energy and then spreads along the channels upon application of the voltage. The liquid also spreads laterally, which I think would make the flow rate control and estimation difficult and highly dependent on the liquid type and the evaporation rate. Device to device variations are also very likely, which I think was not studied by the authors. Considering some laboratory workflows that require flow control, I am not sure how this liquid actuation could be applied to scenarios where multiple liquids or reagents need to flow sequentially (e.g. ELISA) or liquids in multiple channels flow at different rates (e.g. multiplexed assays). To me, protein labeling and viral biosensing applications shown in the manuscript look more interesting than flow control but then I think there are other and more established ways to transfer a liquid from point A to point B in open- or close-channel microfluidics and use electrical actuation for particle manipulation.

Overall: I think it is a compelling and interesting work. But, as an experimental scientist, I do not see an immediate advantage of using this system considering typical applications of microfluidics, especially for point-of-care diagnostics, where using simple and easy-to-use microfluidic systems is of particular interest. On the other side, although I do not feel comfortable to comment on the theory, it seems to be well studied, if it explains a previously unknown phenomenon, this could inspire researchers to apply the method to some specific applications. Finally, because it is strong on the theory, I feel like the manuscript would be more appropriate for a physics journal unless more convincing arguments regarding the motivation and the use-cases are provided.

Response to Reviewers – NCOMMS-21-26399-T

We thank the reviewers for their very thoughtful reading of the manuscript. We found their comments and criticisms valuable in further improving the manuscript. Below, we include the reviewers' comments in italics followed by our responses.

Response to Reviewer #1

Congratulations on reporting a very good proof-of-concept work for low voltage wireless actuation of open microfluidics. I quite enjoyed reading it, to be really honest. Well done!

[Response] We thank Reviewer #1 for their encouraging compliments. As stated, the goal of this work was to provide background theory and proof-of-concept experimentation for manipulating open, micro- and nano-fluidic channels using low-volt, wireless operation. We believe this to be of particular interest for groups performing infrared absorption spectroscopy where low water and channel loss is desired with minimal wire interference under a microscope objective.

Response to Reviewer #2

For the revision, I suggest the authors provide additional context behind the motivation of this work. The authors can elaborate on the significant advantage of their device compared to capillary-driven open microfluidics (see "Suspended microfluidics" published by Casavant et al., PNAS (2013)). From Figure 1D, Figure 3, and Figure 4, there is a clear advantage of this device over capillary-driven microfluidics, so I think authors should definitely make an effort to elaborate on these advantages.

[Response] We appreciate Reviewer #2's feedback and have re-writtent the introduction and conclusion to better highlight the unique advantages of our technology. Please see our responses below as well to address you questions.

From established literature, I would like to direct the authors to the paper, "Electro-actuated valves and self-vented channels enable programmable flow control and monitoring in capillary-driven microfluidics" (Arango et. al., Science Advances (2020)). In this paper, the closed microfluidic device developed by Arango et. al. seems to be comparable to this work (<5V to actuate fluid flow, compact and portable, and fluid control with electric fields). Arango et. al. also claims flow rate significantly higher than flow rate reported for this device in this work (0.5 – 3 nL/s vs 1 pL/min). The difference between this work and work from Arango et. al. is that the channel is open in this work while device from Arango et. al. is closed. Are there any significant advantage/s of open microfluidics in this device? Please elaborate in writing and I think an application that demonstrates the advantage/s of the openness of this device would strengthen this work.

Overall, I find that it was difficult to piece together the significance of this device. For example, it is not clear why this device has to be an open microfluidic device when closed microfluidic devices can perform all these tasks better (see previous paragraph on the work by Arango et. al.). Also, how does this device advance the field of open microfluidics? I suggest conducting major revision to the introduction and conclusion.

[Response] The work presented by Arango et al. was published during the time this work was under its lengthy +1 year review. While we were unable to include that reference initially, we have included

Arango et al. in this most recent draft and a comparison to that work was included into the Introduction where the argument for open-channel microfluidics is made. In summary open-channel systems carry unique advantages to that of enclosed systems. Namely, their lack of component assembly promotes large scale manufacturability and channel dimensions that can be scaled below sub-10 μm —towards single particle, for optimal integration with resonant nano-structures. Further, the full open-channel access enables location-specific surface functionalization, direct sample probing, and the release of unwanted microbubbles that can obstruct enclosed systems (Berthier et al. *Analytical Chemistry* **91**: 8739-8750, 2019). Additionally, without external channel material obstructing the channel viewport, large numerical aperture microscopy can be used for high resolution, real-time imaging and reduced optical loss. The latter is specifically important for infrared (IR) absorption spectroscopy (Figure 3G) in which IR protein resonances compete with the strong absorption of channel material (e.g. polydimethylsiloxane or PDMS) and the ambient saline solution as discussed in my recent commentary (Oh and Altug, *Nature Commun.* **9**:5263, 2018).

1. *What is the advantage of this work compared to capillary-driven open microfluidics?*

[Response] Active open microfluidics, as presented in this work, removes the burden of fluidic operation from the user's handling skills to electronic and algorithmic routines. Our open-channel, active microfluidic platform can be electronically triggered wirelessly from under the microscope for monitoring in real-time (as our included videos demonstrate). As compared to passive platforms, the timing and placement of liquid media depends on the skill of the end-user and every fluid operation consumes increasingly large spatial footprints on the microfluidic platform (e.g. multiple, large mixing structures between each reaction step) (Hessel et al. *Chem. Eng. Science* **60**:2479-2501, 2005) as compared to the active mixing demonstrated within our 100 μm window (Figure 4A).

Furthermore, we demonstrated in this work in Figure 3H-I active particle rejection to filter undesired particles from obstructing the fluidic channel. This would be a challenge with passive capillary as larger, undesired suspended particles in a heterogenous sample that find their way to the channel entrance would obstruct flow. While pre-process procedures such as centrifugation or syringe filtration prior could potentially be implemented, this requires sufficient sample solution and concentrations that such procedures could be implemented rather than drop and go as performed on chip, herein.

2. *Does flow speed change with voltage applied?*

[Response] Yes the voltage applied is expected to affect the flow speed with larger voltages generally resulting in faster flow rates permitting that Joule heating does not evaporate the narrow channels. A balance between the impeding affects of channel evaporation compensated by faster flow rates replenishing the channels will result in an optimal voltage to maximize flow rate. A detailed optimization on the channel flow rate as a function of voltage is reserved for future work, but a discussion on flow rates and its voltage dependence is included in the Supplementary Materials section under "Flow-rate estimations."

3. *Is there significant Joule heating that lead to unwanted droplet evaporation?*

[Response] We did not observe significant Joule heating due to our low-volt operation. As mentioned in the Method's section, our 2–5 μL drops lasted approximately ten minutes under voltage interrogation allowing for sufficient time for mixing, labeling and filtering before evaporation. No signs of bubble formation were observed suggesting little to no hydrolysis present.

4. *Would the channel material affect the flow speed of the device?*

[Response] Yes, the channel material does affect the flow speed. The channel friction factor and hydrophobicity will affect the flow rate of the channel and minimum threshold voltage for operation. Generally, choosing a material or depositing a self-assembled monolayer with a contact angle near the condition for spontaneous capillary flow (e.g. Figure 1D) should render the fastest flow rates. However, the material friction factor can also increase as the contact angle decreases as the sample prefers to interact with the stationary substrate. Therefore, an optimal contact angle should be chosen for a given channel sector angle. A detailed optimization on the channel flow rate as a function of the contact angle is reserved for future work, but a discussion on flow rates and its contact angle dependence was provided in the Supplementary Materials section under “Flow-rate estimations.”

5. In Supplementary Information, please show appropriate steps for the derivation of Equation S4.

[Response] Equation S4 is a well-known relation derived from the Korteweg-Helmholtz (KH) force density. Discussion on this was already provided in two of our provided references (Jones, T. B., *Journal of Electrostatics* **51**: 290-299, 2001), (Mugele et al., *Journal of Physics-Condensed Matter* **17**: R705-R774, 2005). In brief, Equation S4 is the result of a deformable dielectric (sample solution) surrounded by a lower dielectric medium (ambient air) and is actuation due to the tendency for a higher-k dielectric to move towards and fill high electric field regions to reduce the energy of the ensemble. From the KH relation, if we assume the sample is Newtonian (i.e. non-compressible), that we are far from the fringe fields located at the channel edges and thus can eliminate any axial field components, and treat the dielectric permittivity of the sample solution as a delta function at the leading edge of the channel, the KH force density simplifies to Equation S4. We also found the following two references useful if the reviewer desires further reading (Woodson, Herbert H, *Wiley*, Ch. 8, 1968), (Zeng et al. *Lab on a Chip* **4.4**:265-277, 2004).

6. Regarding the calculation of flow rate, did the authors verify if the liquid fully fills the channel cross section when fluid flow is actuated?

[Response] Verification of the channels being fully filled were evident by the resulting salt crystals that remained and filled the channels once the PBS solution dried. As there is no good way to image a diffraction limited fluidic cross-section in real-time, this was deemed sufficient verification that the channels were fully filling the channel cross sections by the salt crystals left behind.

7. Can this device operate when submerged in another liquid (i.e. replacing air with another liquid)? If yes, is there a liquid combination that lowers threshold voltage for OMEF actuation and increases flow rate?

[Response] In theory this device could operate when submerged in another liquid, for example silicon oil. This is advantageous both at mitigating evaporation as well as the surface tensions between silicon oil and water (35-40 dynes/cm) is slightly less than that of water and air (72 dynes/cm) and thus could potentially reduce the operating voltage threshold by $\sqrt{2}$, see Equations 2 and 4. Groups have demonstrated this using liquid dielectrophoresis at much higher voltage devices (see citation Kaler et al. *Biomicrofluidics* **4**:022805, 2010). One concern is the additional drag force imposed by the oil environment which will impede fluid velocity, see the Supplementary Materials section under “Flow-rate estimations.”

8. *Present an application that showcase the advantage/s of the openness of this device.*

[Response] In Figure 3G, we demonstrate low-loss infrared absorption spectroscopy of viral capsids which otherwise would suffer high absorption loss from PDMS channels as discussed in our included references (Oh and Altug, *Nature Commun.* **9**:5263, 2018). Furthermore, each of these operations: mixing, protein labeling, and particle filtration (Figures 3-4) were imaged in real-time using a high magnification 50x microscope objective by simply wirelessly manipulating the fluid under the microscope—a challenge when including top-cover material. We find this to be highly practical for laboratory use enabling high magnification, low-working distance and real-time fluid imaging.

Comments on figures

1. *Figure 1A is inaccurate as “Microfluidics” has characteristic dimensions typically ranging from μm to mm.*

[Response] We are confused by this comment as Figure 1A does in fact represent Microfluidics as spanning characteristic dimensions from $10\ \mu\text{m} - 1\ \text{mm}$ (x-axis: 10^{-5} - 10^{-3} m). Enclosed microfluidic channels that are completely confined in all direction by $< 10\ \mu\text{m}$ are extremely rare and thus does not represent the normal operating regime of microfluidics as Figure 1A was intending to demonstrate.

2. *Please include label for each part of the device in Figure 1B.*

[Response] We are unsure what components the reviewer is desiring for this schematic as Figure 1B was provided to define the key symbols used in the subsequent equations derived in the main text. If the reviewer is desiring a figure of the chip device we kindly would direct the reviewer to Figure S1 in the supplementary materials section which shows the actual chip, location of the droplet reservoir, and working electrodes.

3. *Please include an intermediate frame for Figure 3B-C, Figure 3E-F, and Figure 4D as it is hard to tell what is happening with the current frames presented.*

[Response] The videos files for each of these figure sets were already provided to clearly demonstrate the real-time channel progression of these operations. Therefore, due to limited figure space, the multiple intermediate frames were omitted.

Response to Reviewer #3

This paper represents an innovative design of open fluidic microsystem that addresses the fundamental challenge of surface tension in open-channel design. Using a combination of nano-RF field focusing and a lower surface energy electrode geometry, they demonstrated OMEF actuation using a wireless, digital RF power signal. They show the actuation of microchannels as narrow as $1\ \mu\text{m}$ using voltages as low as $0.5\ \text{VRMS}$ for both dielectric and physiological liquids. They further demonstrated the practical fluidic applications including open mixing, lateral-flow protein labeling, filtration, and viral transport for efficient biosensing. This system is coupled with resonant wireless power transfer for remote actuation, with smartphone-driven fluidics presented to fully showcase the potential of this open-chip technology.

Though thorough experimental observations are performed, there are several fundamental aspects which the authors did not address properly. I may recommend publishing this work on ‘Nature

Communication' if the authors can address my following comments and overhaul the manuscript thoroughly in terms of additional experiments and physics.

1. In equation 2, for a planarized open-channel chip, what is the limiting condition of the electrodes spacing for maximum E_t as it doesn't show any direct dependence on 'g'.

[Response] The dependence of E_t on "g" was modeled using the electrostatics model in COMSOL, see the "Modeling" section in the Supporting Materials. Fabrication constraints and electrical shorting are the fundamental limiting factors on the electrode spacing, "g". It is quite difficult to generate millimeter long regions with planarized nanometer size electrode gaps that do not present any shorts or pin holes (Barik et al., *Nano letters* **16.10**: 6317-6324, 2016). As the electrode spacing, "g" is reduced not presenting any shorts is a fundamental limit and dielectric breakdown occurs at a lower operating voltage (see Figure S2). While the transverse electric field, E_t , does increase as "g" is reduced, the extent that the strong field extends into the fluidic channel, R , reduces, see Modeling section in the Supporting Information and Figure S3 which reduces the spatially averaged actuating force over the channel cross-section.

In case of vertical stacked electrodes, what is the deciding factor for choosing 'g = 20 nm'?

[Response] Similar to the response to question 1, the 20 nm electrode gap was chosen out of practicality. Values lower than 20 nm resulted in poor device yield due to electrode shorting while values larger than 20 nm resulted in larger, unwanted operating voltages. A comment regarding this was added to the Supplementary materials.

2. *What is the passivation layer of 5 nm signifies here as both the g and t is considered here as same?*

[Response] We found this question a bit tricky to understand as to what is being asked but will give our best shot. Due to the effective oxide thickness $t \ll g$, the electrode gap could be modeled using just "g." Please see the "Modeling" section in the Supporting Materials. The effective oxide thickness of the $t = 5$ nm TiO₂ layer coated to normalize the contact angle compares to a ~ 0.6 nm layer of Al₂O₃. This sits on top of a 20 nm Al₂O₃ gap layer in which its combined effective thickness is within one standard deviation of the fabrication process (Table S1).

This value "g=20 nm Al₂O₃" was then used to model E_t for comparing the dielectric fluid condition (Equation 2) and was used as the value "t" in Equation 4. In our current stacked electrode configuration, the electrode gap, "g," happens to also be the passivation layer thickness, "t" used for the conductive liquid operating threshold (Equation 4). However, generally this is not the case. Consider Figure 1B where the electrode gap, "g," is now different from the passivation layer thickness, "t." Therefore, a configuration such as Figure 1B would allow for the designer to optimize the operating conditions for Equation 2 or 4 independently by changing "g" or "t," respectively.

3. *Is there any dependence of projection length and width on the electrode geometry (like a and R)?*

[Response] Yes, the channel width correlates with "R" as "R" defines the radius of the channel sector (Figure 1B and 1D). However, the observed width of the channel will also depend on the angle at which the channel is viewed from. For example, as the sidewall angle, a , is reduced, less of the channel width can be viewed from above due to more of the channel beneath the over-hang of the top electrode.

The projection length also depends on these factors and we will direct the reviewer to the “Flow-rate estimations” section in the Supporting Materials. There the dependence on the channel length is discussed and its geometric dependence.

4. *How does the energy cost in Fig. 1D increase with α as fabricating $\alpha = 0$ and p is easier than $\alpha \neq 0$ and p , so it should follow a parabolic trend here instead of the existing one.*

[Response] We found this question as written a bit confusing to understand. In terms of fabrication, a sidewall angle of $\alpha = 0$ physically does not make sense as there would be no space between the electrodes. If the reviewer is suggesting using a parallel plate top electrode, this then is no longer an open microfluidic channel in which the motivation for open channel microfluidics is discussed in the introduction section. Please see our response to question 5 below for more clarification on figure 1D.

5. *What is the reason of using $\alpha \sim 71^\circ$ in this experiment, as in Fig. 1D, a close to 90° gives the optimum setting for getting less driving force to drive the liquid and less energy cost?*

In connection to this, how does the author use Pos. PR to selectively coat with an obtuse side wall angle to pattern the top electrode in Fig. S1? Please explain it properly.

[Response] We believe the reviewer may have misunderstood Figure 1D as the location where the two curves overlap around 90° is coincidence and has no meaning, i.e. each curve has a different y-axis. Instead, Figure 1D is intended to show that as the sidewall angle, α , is reduced we achieve a two-fold advantage. We get a reduction in the surface energy cost for actuation (black curve reduces) and subsequently an increase in the electromechanical force on the liquid (red curve increases). Once the surface energy cost reaches zero (at the gray dashed line), spontaneous capillary action will occur and actuate the channel. Therefore, the optimum angle is one as close to the spontaneous capillary line (gray dashed line) such that actuation can be triggered using a low-volt RF signal. Then the operating voltage will be at the lowest possible value to trigger spontaneous capillary action for liquid actuation.

In order to generate an obtuse sidewall angle several fabrication techniques could be implemented. A negative tone resist could be used instead of the POS. PR which gives an obtuse sidewall angle. Further, patterning the top metal using an etch back process rather than a lift-off process could be implemented to generate an obtuse sidewall angle as well. An obtuse sidewall would improve the optical access of the channel from above but would increase the operating voltage for actuation.

6. *At what frequency the average threshold voltages in Fig. 2B are sufficient to form a microfluidic channel for both DI water and PBS? What is the limiting setting for V_{RMS} and frequency for avoiding Joule heating here?*

[Response] The general operating frequency for the average threshold voltages in Figure 2B was 100 kHz, see Method's section. Depending on the operating frequency, the dominant mode of actuation on the liquid body will change. If the frequency is larger than the cross-over frequency, see Supporting Materials section “Conductive to Dielectric Cross-over Frequency” than dielectric body forces dominate (Equation 2) and thus a response similar to the blue curve in Figure 2B is expected. If the frequency is lower than the cross-over frequency, than electrowetting forces dominate (Equation 4) and thus a response similar to the gold curve in Figure 2B is expected.

Part of the novelty of this work was demonstrating that regardless if you are operating with a frequency and liquid solution conductivity that puts you in the dielectric body force regime (Equation

2) or electrowetting regime (Equation 4) that you could achieve low-volt actuation by tailoring your electrode geometry (specifically via the sidewall angle, α , in this work).

The advantage this gives then is flexibility when optimizing one's circuit and choosing the target operating frequency. Figure 2C was intended to show this by designing a resonant circuit, Equation S1, the operating frequencies could be adjusted as needed to reduce the operating VRMS voltages further, Figure 2D.

The goal was to keep all operating voltages below that of standard digital logic (3.5 VRMS). While Joule heating is inevitable, our low-volt operation helps mitigate these effects. At our operation (≥ 100 kHz), our 2-5 μl drop reservoirs would last approximately 10 minutes which was sufficient for our experiment before evaporation, see Method's section.

Further circuit optimization to reduce power consumption and Joule heating is reserved for future work, however, within the scope of this work, the geometric gains implemented herein significantly reduced operating voltage, see cited work and Figure 1C in main text where operating voltage from previous groups ~ 100 VRMS.

7. Is there any influence of DEP force on the intensification of voltage gain in Fig. 2C for Resonant circuit at 1 MHz frequency? There is highly likely possibility of having the effect of DEP force on the gain due to the electrode geometry. Please justify the reason of getting a high gain at 105-106 Hz in Fig. 2C.

[Response] We found this question a bit tricky to understand as to what is being asked but will give our best shot. The inductor, see Methods section and Supporting Materials, in resonance with the device capacitance, Equation S1, provides the large voltage gain seen in Figure 2C at ~ 400 kHz, i.e. a resonant LCR circuit. This was designed as such. It is expected the DEP force does become stronger when using the resonant tank circuit as demonstrated in Figure 2C-D due to an increase in the electric field gradient at the channel entrance, Equation S24.

Response to Reviewer #4

The manuscript by Ertsgaard et al. proposes a liquid and particle actuation mechanism in microfluidics using an open-channel architecture and wireless electrical power delivery. The manuscript is well-written, has nice and clear figures, and looks complete with extensive theoretical background, modeling, and experimental results.

[Response] We thank the reviewer for their kind comments regarding the quality of the work and its presentation.

1- I was not familiar with the term "Open-micro-electrofluidics (OMEF)" and I could not find it on the internet. I think it is a term coined by the authors to also include open-channel EWOD and open-channel DEP. To me, this worked looked like a combination of surface-tension driven liquid pumping (e.g. Gosh et al. Lab Chip, 2014), open-channel capillary systems (e.g. Berthier et al. Anal. Chem. 2019) and digitally controlling the flow of continuous liquids using electrowetting (e.g. virtual electrowetting channels, Banerjee et al. Lab Chip, 2012). But overall, I think it is a novel implementation by its own.

[Response] Indeed, the novelty of this work was derived by realizing the physics of electrowetting and dielectric body forces both benefit from optimizing the electrode geometries. Specifically, in this work, we found that by creating a $< 180^\circ$ angle between the working electrodes, a two-fold advantage could be found: reduced surface tension and better confined radio frequency power (Figure 1D). This

enabled very low-power and wireless operation (Figure 2) as compared to previously before (Figure 1A). We coined the term OMEF to clearly highlight both the tightly confined and open-channel attributes that together are currently very difficult to electrically modulate using reasonable voltage (Figure 1A) until now.

2- Open-channel microfluidics clearly bring some advantages for fabrication (no sealing required) and operation of devices (removing air bubbles, easily merging multiple liquids). But to me the real benefit is to be able to integrate bioreagents to microchannels because it's often the case that sealing/closing step requires high temperature or the use glues or solvents that are not compatible with bioreagents. Unfortunately, the authors do not talk about this advantage and the microfluidic architecture does not look compatible with pre-integration of reagents. Although the final sealing step is omitted in the open-channel architecture, the fabrication process looks still complicated and demanding (e.g. the use of ALD) compared to other open-channel implementations. It was difficult for me to see the motivation behind open-channel architecture for the use-cases demonstrated in the manuscript.

[Response] Well we would be interested in what level of pre-integration of reagents the reviewer is considering, we do mention in the Introduction section that the open-channel platform enables location specific surface functionalization that can be pre-integrated onto the device or even full submersion of the device for large scale, easy surface modification due to no top covers.

We appreciate that “simplicity” of the fabrication process can be a relative term. Our definition considers the number of steps, likelihood these steps can result in error, and finally, how easily they can be scaled up for manufacturing. Since the fabrication of this chip uses routine CMOS processes (e.g. photolithography, metal evaporation, and ALD) that can all be automated, we believe the manufacturability to be more robust and “simpler” as compared to classical microfluidics which must integrate tubes, valves, interconnects, and water-tight seals in the assembly process—often by hand, which adds to the complexity and cost of manufacturing.

In terms of user error, enclosed systems are also susceptible to microbubble formation as you mentioned, which can impede flow and disrupt experiments. This is compared to open-channel systems in which they can freely escape. As such, proper equipment and/or expertise (e.g., degassing, efficient valve switching, etc.) must be supplied by the end user, with more chance for error. This is only exacerbated as multiple solutions and flow steps are introduced. Our technology removes the burden of assembly and finesse from its operators by instead using controlled CMOS-fabricated electrodes to automate and perform open-channel fluidic actions. Many of these items were better emphasized in the introduction and conclusion section.

3- Low-power actuation and smartphone compatibility are great features. But it is hard to justify without knowing the target application. If it is for point-of-care diagnostics, not being able to integrate reagents would be a bottleneck I think (my previous comment). If it is more for a research platform, then the smartphone compatibility may not be as important. The manuscript demonstrates some nice use cases, but again it is difficult for me to justify why those applications were chosen and in what context.

[Response] The capability of low-power helps reduce evaporation rates and promotes wireless coupling of RF power. The wireless aspect was presented to offer an external “hardware” free platform, i.e. no tubing, top-cover, or wire connections obstructing the channel viewport. The advantage this offers is the capability for low working distance, large numerical aperture microscopy for high resolution, real-time imaging as all of these channels were imaged using a 50x objective (see Methods). Further, confined microchannels with no top-cover reduces optical loss that is specifically important for infrared (IR) absorption spectroscopy biosensing. IR protein resonances compete with the strong absorption of channel material (e.g. polydimethylsiloxane or PDMS) and the ambient

saline solution and thus direct optical access is greatly preferred (Oh and Altug, *Nature Commun.* 9:5263, 2018). This was better emphasized in the main text. The demonstration of the smartphone compatibility was to showcase the other advantage for hardware free, low-power operation and that is portability for point-of-care applications. This was demonstrated as a proof-of-concept to inspire the future evolution of this work.

4- Although being impressive and convincing, I think the submitted work does not enable a real flow control mechanism for capillary microfluidics. As can be seen in the videos, the liquid advances in the channels by capillarity without applying any energy and then spreads along the channels upon application of the voltage. The liquid also spreads laterally, which I think would make the flow rate control and estimation difficult and highly dependent on the liquid type and the evaporation rate. Device to device variations are also very likely, which I think was not studied by the authors. Considering some laboratory workflows that require flow control, I am not sure how this liquid actuation could be applied to scenarios where multiple liquids or reagents need to flow sequentially (e.g. ELISA) or liquids in multiple channels flow at different rates (e.g. multiplexed assays). To me, protein labeling and viral biosensing applications shown in the manuscript look more interesting than flow control but then I think there are other and more established ways to transfer a liquid from point A to point B in open- or close-channel microfluidics and use electrical actuation for particle manipulation.

[Response] We appreciate the reviewer's comments regarding flow rates and device repeatability as these are important issues to consider. We did provide discussion on both in the Supplementary Materials section (see "Flow-rate estimations," "Electrode edge, contact angle, and capacitance characterization" section, and Tables S1-S4) as well as the "Statistics" section of our Methods section. Indeed, characterization of flow rates is addressed but further characterization is beyond the scope of this work.

Overall: I think it is a compelling and interesting work. But, as an experimental scientist, I do not see an immediate advantage of using this system considering typical applications of microfluidics, especially for point-of-care diagnostics, where using simple and easy-to-use microfluidic systems is of particular interest. On the other side, although I do not feel comfortable to comment on the theory, it seems to be well studied, if it explains a previously unknown phenomenon, this could inspire researchers to apply the method to some specific applications. Finally, because it is strong on the theory, I feel like the manuscript would be more appropriate for a physics journal unless more convincing arguments regarding the motivation and the use-cases are provided.

[Response] Please see the revised introduction in which we more clearly outlined the advantages of this work. As mentioned above, FTIR spectroscopy for biosensing (Figure 3G) greatly benefits from our design where we remove the top-channel material and reduce the aqueous channel volume which contributes large absorption losses. Practically speaking, active open microfluidics, as presented in this work, removes the burden of fluidic operation from the user's handling skills to electronic and algorithmic routines. As compared to passive platforms, the timing and placement of liquid media depends on the skill of the end-user and every fluid operation consumes increasingly large spatial footprints on the microfluidic platform (e.g. multiple, large mixing structures between each reaction step)(Hessel et al. *Chem. Eng. Science* 60:2479-2501, 2005). Instead, we demonstrated active mixing within a 100 μm window (Figure 4A). We believe this monolithic platform that can actively filter large particles from the channel regions, perform sample mixing and protein labeling with FTIR sensing, wirelessly under a microscope to be of great value to the scientific community.

REVIEWERS' COMMENTS

Reviewer #2 (Remarks to the Author):

The authors has responded to my comments excellently. No further revision is needed.

Reviewer #3 (Remarks to the Author):

I am satisfied with the revised manuscript.

Reviewer #4 (Remarks to the Author):

I thank authors for taking my comments into consideration and adapting the manuscript accordingly. I found the their response satisfactory. The work was already technically sound and the revised version of the manuscript is now substantially improved with the addition of more discussion on the applications and the advantages of the proposed approach. I recommend its publication.